# Where are patients missed in the tuberculosis diagnostic cascade? A prospective cohort study in Ghana

**Joyce B. Der**[1,2]*, **Daniel Grint**[1], **Clement T. Narh**[2,3], **Frank Bonsu**[4], **Alison D. Grant**[1,5,6]

**1** TB Centre, London School of Hygiene & Tropical Medicine, London, United Kingdom, **2** Department of Epidemiology and Biostatistics, School of Public Health, University of Health and Allied Sciences, Hohoe, Ghana, **3** Institute for Medical Biostatistics, Epidemiology and Informatics, University Medical Centre of the Johannes Gutenberg – University Mainz, Mainz, Germany, **4** Department of Disease Control and Prevention, National TB Control Program, Ghana Health Service, Accra, Ghana, **5** Africa Health Research Institute, School of Nursing and Public Health, University of KwaZulu-Natal, Durban, South Africa, **6** School of Public Health, University of the Witwatersrand, Johannesburg, South Africa

* joyce.der@lshtm.ac.uk

**Data Availability Statement:** The data underlying this study contain sensitive patient information and cannot be made publicly available. The data have been deposited with the London School of Hygiene

## Abstract

### Background

Ghana's national prevalence survey showed higher than expected tuberculosis (TB) prevalence, indicating that many people with TB are not identified and treated. This study aimed to identify gaps in the TB diagnostic cascade prior to starting treatment.

### Methods

A prospective cohort study was conducted in urban and rural health facilities in south-east Ghana. Consecutive patients routinely identified as needing a TB test were followed up for two months to find out if sputum was submitted and/or treatment started. The causal effect of health facility location on submitting sputum was assessed before risk factors were investigated using logistic regression.

### Results

A total of 428 persons (mean age 48 years, 67.3% female) were recruited, 285 (66.6%) from urban and 143 (33.4%) from rural facilities. Of 410 (96%) individuals followed up, 290 (70.7%) submitted sputum, among which 27 (14.1%) had a positive result and started treatment. Among those who visited an urban facility, 245/267(91.8%) submitted sputum, compared to 45/143 (31.5%) who visited a rural facility. Participants recruited at the urban facility were far more likely to submit a sputum sample (odds ratio (OR) 24.24, 95%CI 13.84–42.51). After adjustment for confounding, there was still a strong association between attending the urban facility and submitting sputum (adjusted OR (aOR) 9.52, 95%CI 3.87–23.40). Travel distance of >10 km to the laboratory was the strongest predictor of not submitting sputum (aOR 0.12, 95%CI 0.05–0.33).

and Tropical Medicine's Data Compass and are accessible using the following DOI: https://doi.org/10.17037/DATA.00001529. Interested researchers wishing to request access to the data can do by clicking the "request access" button to fill a form which will be automatically submitted to the research data manager at London School of Hygiene and Tropical Medicine and to the first author. In addition, requests can be sent to researchdatamanagement@lshtm.ac.uk.

**Funding:** The author(s) received no specific funding for this work.

**Competing interests:** The authors have declared that no competing interest exist.

## Conclusion

The majority of presumptive TB patients attending a rural health facility did not submit sputum for testing, mainly due to the long travel distance to the laboratory. Bridging this gap in the diagnostic cascade may improve case detection.

## Introduction

Ghana is one of the tuberculosis (TB) and human immunodeficiency virus (HIV) high burden countries according to the World Health Organization (WHO) [1]. Ghana's TB case notification rate has declined from 56/100,000 in 2014 [2] to 52/100,000 in 2017[3] compared to an estimated incidence of 152/100,000 based on the 2013 national TB prevalence survey [4]. Ghana's national TB prevalence survey also highlighted weaknesses in the care cascade where of persons with prolonged cough who visited a health facility, only 25% submitted sputum for testing [4].

In Ghana, TB diagnosis and treatment is mainly done at government health facilities. Until recently, sputum smear microscopy was the main diagnostic method, but Xpert MTB/RIF has now been introduced with 105 GeneXpert machines installed nationwide by 2017 [3]. Most diagnostic laboratories are located at secondary and tertiary levels of care but not at peripheral levels such as health centres mostly in rural areas. Therefore, a person with symptoms of TB might be identified at a health centre but will have to travel themselves to the district hospital for a sputum test to confirm the diagnosis. This creates an obstacle to diagnosis and treatment and potential loss within the cascade of TB care.

There is a wealth of literature on health facility contribution to delayed or missed diagnosis of TB and pre-treatment loss-to-follow up [5–9]; however, most are cross-sectional studies involving TB patients already on treatment, thus excluding those who never started treatment, or retrospective reviews of secondary data [10]. To better understand losses from the care cascade, we conducted a prospective observational cohort study to identify where and when potential TB patients are missed in the diagnostic cascade. We hypothesized that distance to the laboratory was a key determinant of whether sputum was submitted or not, therefore, persons with symptoms suggestive of TB (presumptive TB patients) were less likely to submit a sputum if they attended a rural health facility without a laboratory compared to those attending an urban facility with a co-located laboratory. Our aim was to determine if persons asked to submit sputum for testing for TB submitted sputum, and whether this was associated with urban vs. rural location of the facility; also whether they received test results, and if the results were positive, whether they started TB treatment.

## Materials and methods

### Study setting

The study was conducted in Ketu South Municipality in the Volta region of Ghana, which shares boundaries with the Republic of Togo. The municipality has one government hospital and eight health centres. In 2017, the municipality notified 290 TB cases out of a target of 534 estimated based on the 2013 national TB prevalence survey. In 2018, it notified 172 TB cases out of a target of 546, indicating a decreased case detection rate from 53.8% in 2017 to 31.5% in 2018 [11]. There is only one TB diagnostic laboratory in the municipality, located at the municipal hospital. All other health facilities in the municipality refer patients with symptoms

of TB to the municipal hospital for testing. The study was thus conducted in the municipal hospital, and four health centres without co-located TB diagnostic facilities in rural areas located 10 to 20 km from the municipal hospital. The four health centres were selected based on high outpatient department (OPD) attendance and to represent the different health demarcated sub-municipalities.

## Study design

This was a prospective observational cohort study among adults aged ≥18 years with or without symptoms suggesting TB, self-presenting at the health facility, who were identified routinely by a health worker as needing TB investigation and given a laboratory request to do a sputum test or referred to the municipal hospital.

**Sampling strategy.** We consecutively invited eligible patients from the selected health facilities from May 2018 until the target study population had been enrolled in February 2019.

## Data collection

At enrolment, trained research assistants collected baseline information on socio-demographic characteristics, symptoms of illness, health histories and date the sputum test was requested, using a standardised questionnaire. The questionnaire was pretested at the municipal hospital in a population similar to the study population. Following pretesting, revisions were made to improve comprehensibility and understanding. Participants were followed up for two months after the test request, via two-weekly phone calls to find out if sputum had been submitted, when and where it was submitted and the result of the test; and among those with a positive test result, to determine whether the patient had started TB treatment. Those without phones were visited at their homes once monthly for two months. Geographic positioning coordinates (GPS) of participant households and health facilities attended were collected. Coordinates were stored separately from participant information to ensure confidentiality. All data were collected electronically using Open Data Kit (ODK) and uploaded onto a secured server hosted by the London School of Hygiene & Tropical Medicine.

## Sample size

Assuming 49% of participants attending an urban facility would submit sputum [7], we calculated that a sample size of 414 would provide 80% power to detect a difference of 15% in the proportion of participants who submitted a sputum in an urban versus a rural facility with alpha set to 5% and allowing 10% loss to follow up.

## Measures and definitions

TB symptoms were defined as the four cardinal symptoms of TB (cough, fever, weight loss and night sweats) and/or other TB symptoms (chest pain, coughing up blood, tiredness, shortness of breath and/or lump in neck). Rural and urban residence was defined based on the municipal health directorate categorization. Socioeconomic status was generated using asset scores which included 33 items, based on methods used by demographic and health surveys [12]. Participants' attitudes of stigma towards people with TB were measured using a tool developed by Van Rie et al, based on 12 questions measured on a five-point Likert scale, with 1 being lowest score for stigma and 5 being the highest [13]. The mean score across the 12 questions was calculated and dichotomized to represent high perceived or low perceived stigma. Distance from a participant's residence to the TB diagnostic facility was generated from GPS coordinates. Body mass index (BMI) was categorized according to WHO guidelines [14]. A Karnofsky

score was estimated as a measure of illness severity [15]. Diabetes and HIV status were defined based on participant self-report.

**Primary outcome.** This was the proportion of participants who submitted a sputum sample.

### Data management and statistical analysis

Primary analysis assessed the causal effect of type of health facility attended (urban versus rural) on the probability of submitting sputum, using logistic regression. Any factor that changed the odds ratio by approximately 10% in bivariable analysis was considered a potential confounder and adjusted for in a final multivariable model. Kaplan-Meier analysis was used to compare the time to submitting sputum between urban and rural facility attendees. Predictive modelling was employed to assess risk factors associated with submitting sputum using logistic regression. Variables with likelihood ratio p-value <0.2 in univariable analysis were included in a multivariable model. Data were analysed using Stata v15 (Stata Corp, College Station TX, USA).

### Ethical considerations

Ethical approval was obtained from the Ghana Health Service Ethics Review Committee and London School of Hygiene & Tropical Medicine Ethics Committee. Written informed consent was obtained from literate participants and for those who could not read and write, consent was documented with a thumbprint in the presence of a literate witness.

## Results

A total of 468 persons were approached: of this number 437 (93.4%) were eligible, 428/437 (97.9%) consented and were recruited (285 from urban facility and 143 from rural facilities). The main reason for non-eligibility was being <18 years (28, 90.3%). Of the 428 recruited, 397 (92.8%) completed follow-up for two months, 13 (3.0%) died at some point in time within the two-month follow-up period and 17 (4.0%) were lost to follow up (Fig 1).

### Baseline characteristics of study participants

Among 428 participants, the mean age was 48 years (standard deviation [SD]18.8) and this was similar among urban and rural facilities (Table 1). The majority of participants (288, 67.3%) were female and most (237, 55.4%) had attained primary level education. The median distance between participants' home and the laboratory was 17.3 km (interquartile range [IQR]16.7–19.4) for rural facility attendees and 2.6 km (IQR 1.3–10.0) for urban facility attendees. Most participants attending rural facilities were in the lowest socioeconomic tertile (58, 40.6%) compared to those attending urban facilities (85, 29.8%) and more than half (255, 59.6%) of study participants had a high perception of TB related stigma.

### Cascade of care

Of 410 presumptive TB patients asked to do a sputum test who were followed up, 290 (70.7%) submitted sputum for the test; 194/290 (66.9%) received a test result; 27/194 (13.9%) had a positive test result and all 27 started TB treatment (Fig 2). Sixteen additional patients started treatment based on chest radiograph findings. Stratifying by type of health facility attended, 245/267 (91.8%) submitted sputum among those attending the urban facility and 45/143 (31.5%) among those attending rural health facilities (Fig 2).

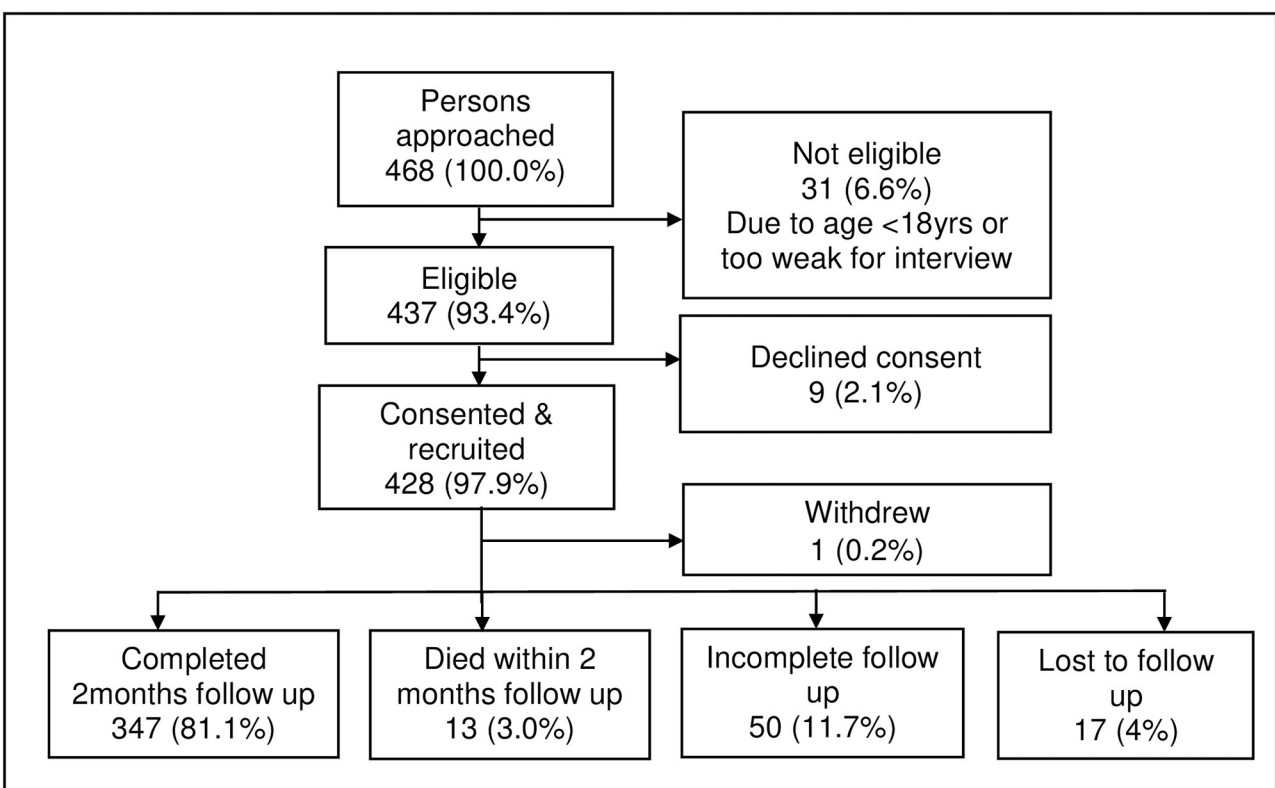

**Fig 1. Flowchart of study recruitment and follow up of people requested to give sputum for TB investigation in Ketu South Municipality, Ghana.**

### Time to submitting sputum among rural and urban health facility attendees

In Kaplan-Meier analysis, there was strong evidence for a difference in time to submitting sputum by location of health facility, where urban facility attendees submitted sputum earlier than rural facility attendees (P<0.001) (Fig 3). Of urban facility attendees, 229 (80.4%) submitted sputum on the day the test was requested, 10 (3.5%) submitted within 7 days and 6 (2.1%) after 14 days. However, among rural facility attendees, only 6 (4.2%) submitted sputum on the day the test was requested, 18 (12.6%) within 7 days and 21 (14.7%) after 14 days (Fig 3).

### Reasons for not submitting sputum

The main reasons given by participants for not submitting sputum were being too busy to go to the laboratory (58/116, 50.0%), feeling well so not seeing the need to do the test (19, 16.4%), having no money to travel to do the test (17, 14.7%) and being unable to produce sputum (15, 12.9%).

### Association between type of health facility attended and submitting sputum

In causal analysis, participants recruited from the urban health facility had a much higher odds (unadjusted odds ratio [OR] 24.24, 95% CI 13.84–42.51) of submitting sputum compared to those recruited at a rural health facility. Adjusting for distance from participants' residence to the laboratory resulted in the biggest reduction in the odds of submitting sputum, comparing

**Table 1. Baseline characteristics of participants (people requested to give sputum for TB investigation) in Ketu South Municipality, Ghana.**

| Characteristics | Variable | Urban health facility, n = 285 | Rural health facility, n = 143 | Total, N = 428 | P-value |
|---|---|---|---|---|---|
| | | n (%) | n (%) | n (%) | |
| Age (years) | Mean ± SD | 47.6±17.9 | 48.4±20.4 | 47.9±18.8 | 0.901 |
| Gender | Male | 99 (34.7) | 41 (28.7) | 140 (32.7) | 0.207 |
| | Female | 186 (65.3) | 102 (71.3) | 288 (67.3) | |
| Educational level | No formal education | 70 (24.6) | 34 (23.8) | 104 (24.3) | 0.982 |
| | Primary/JHS | 157 (55.1) | 80 (55.9) | 237 (55.4) | |
| | Secondary/Tertiary | 58 (20.4) | 29 (20.3) | 87 (20.3) | |
| Place of residence | Urban | 205 (71.9) | 1 (0.7) | 206 (48.1) | <0.001 |
| | Rural | 80 (28.1) | 142 (99.3) | 222 (51.9) | |
| Distance to laboratory (km) | Median (IQR) | 2.6 (1.3–10.0) | 17.3 (16.7–19.4) | 10.0 (1.6–17.3) | <0.001 |
| Socioeconomic status (tertiles) | High | 114 (40.0) | 28 (19.6) | 142 (33.2) | <0.001 |
| | Middle | 86 (30.2) | 57 (39.9) | 143 (33.4) | |
| | Low | 85 (29.8) | 58 (40.6) | 143 (33.4) | |
| TB symptoms (yes) | Cough | 279 (97.9) | 142 (99.3) | 421 (98.4) | 0.279 |
| | Fever | 204 (71.6) | 115 (80.4) | 319 (74.5) | 0.048 |
| | Night sweat | 70 (24.6) | 24 (16.8) | 94 (22.0) | 0.067 |
| | Weight loss | 79 (27.7) | 22 (15.4) | 101 (23.6) | 0.005 |
| Symptoms duration (days) | Median (IQR) | 19 (7–31) | 7 (7–14) | 14 (7–28) | <0.001 |
| HIV status | Positive | 32 (11.2) | 0 (0.0) | 32 (7.5) | <0.001 |
| | Negative | 72 (25.3) | 17 (11.9) | 89 (20.8) | |
| | Don't know | 181 (63.5) | 126 (88.1) | 307 (71.7) | |
| Diabetes | No | 273 (95.8) | 142 (99.3) | 415 (97.0) | 0.046 |
| | Yes | 12 (4.2) | 1 (0.7) | 13 (3.0) | |
| BMI (kg/m2) | Median (IQR) | 24.6 (19.7–34.1) | 25.1 (22.0–30.3) | 24.8 (20.3–32.6) | 0.077 |
| Severity of illness (Karnofsky score) | Mean ± SD | 81.2±10.8 | 86.1±6.4 | 82.9±9.8 | <0.001 |
| Stigma | Low perceived stigma | 138 (48.4) | 35 (24.5) | 173 (40.4) | <0.001 |
| | High perceived stigma | 147 (51.6) | 108 (75.5) | 255 (59.6) | |

N = total number, n = number within facility, SD = standard deviation, IQR = interquartile range, BMI = body mass index, TB = tuberculosis, HIV = human immunodeficiency virus, JHS (13–15 years) = junior high school

urban and rural facility attendees (adjusted OR [aOR] 12.98, 95% CI 5.95–28.30) (Table 2). After adjusting for all other confounders, in the final adjusted model, a strong association between attending the urban versus rural health facilities and submitting sputum remained (aOR 9.52, 95% CI 3.87–23.40) (Table 2).

## Factors associated with submitting sputum among study participants attending an urban or rural health facility

In univariable analysis (Table 3), rural versus urban residence (OR 0.08, 95% CI 0.05–0.15); longer travel distance to the laboratory (OR 0.06, 95% CI 0.03–0.11 for 10–20 km and OR 0.17, 95% CI 0.07–0.41 for >20 km versus <10 km); and high versus low perception of TB-related stigma (OR 0.39, 95% CI 0.24–0.62) were associated with a lower odds of submitting sputum. Prior TB treatment (OR 4.00, 95%CI 1.19–13.460); more reported symptoms (OR 2.87, 95% CI 1.47–5.59 for >4 versus 1–2), longer symptom duration (OR 2.20, 95% CI 1.39–3.50 for >14 versus ≤14 days); and visit to at least one care providers prior to current clinic visit (OR 3.18, 95% CI 1.88–5.10) were predictors of submitting sputum.

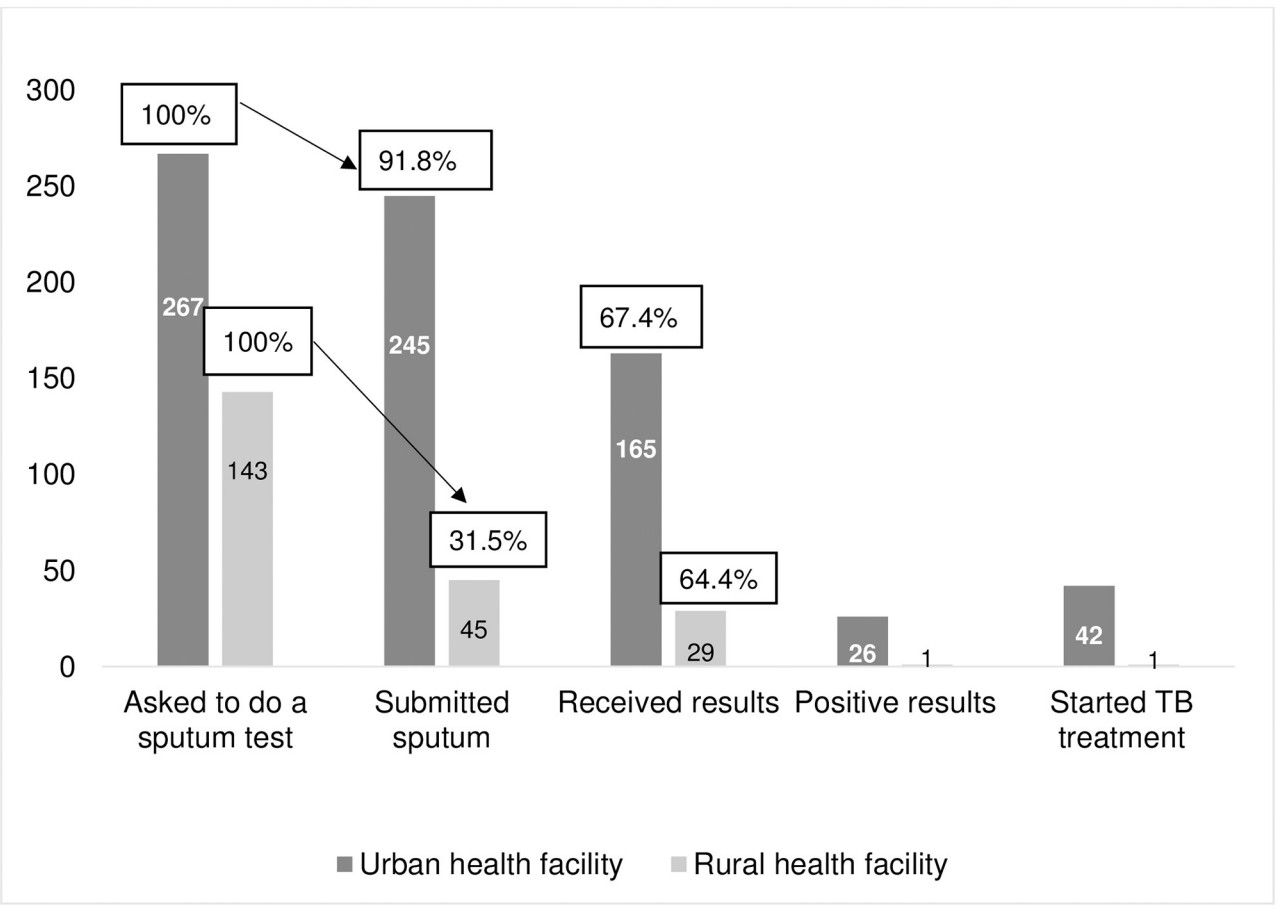

**Fig 2. Cascade of TB care among people requested to give sputum for TB investigation in Ketu South Municipality, Ghana.**

In predictive multivariable analysis, longer travel distance to the laboratory and high perception of TB-related stigma remained associated with a lower odds of submitting sputum (Table 3).

## Discussion

In south-east Ghana, more than a quarter of presumptive TB patients never submitted a sputum, and non-submission of sputum was most strongly associated with longer distance to the laboratory. Studies from Zimbabwe, India and Tanzania have also reported high pre-diagnosis loss-to-follow, 25%, 30.4% and 44% respectively [16–18], and some studies, including one from Ghana, showed that attending rural health clinics and long travel distance were risk factors for delay or pre-diagnosis loss to follow-up [7, 10, 17, 19]. In contrast, pre-diagnosis attrition was lower in South Africa and China (5% and 11% respectively) [20, 21], perhaps because in South Africa, sputum specimens are transported free to a central laboratory for diagnosis.

TB-related stigma was independently associated with non-submission of sputum, consistent with findings from a study in India [22]. Several studies have reported high perception of TB-associated stigma [23–27], and specific actions are needed to counter this [28, 29].

An encouraging finding was that all patients with a positive TB result on their sputum were promptly put on treatment. This contrasts with a study of routine data from a regional hospital

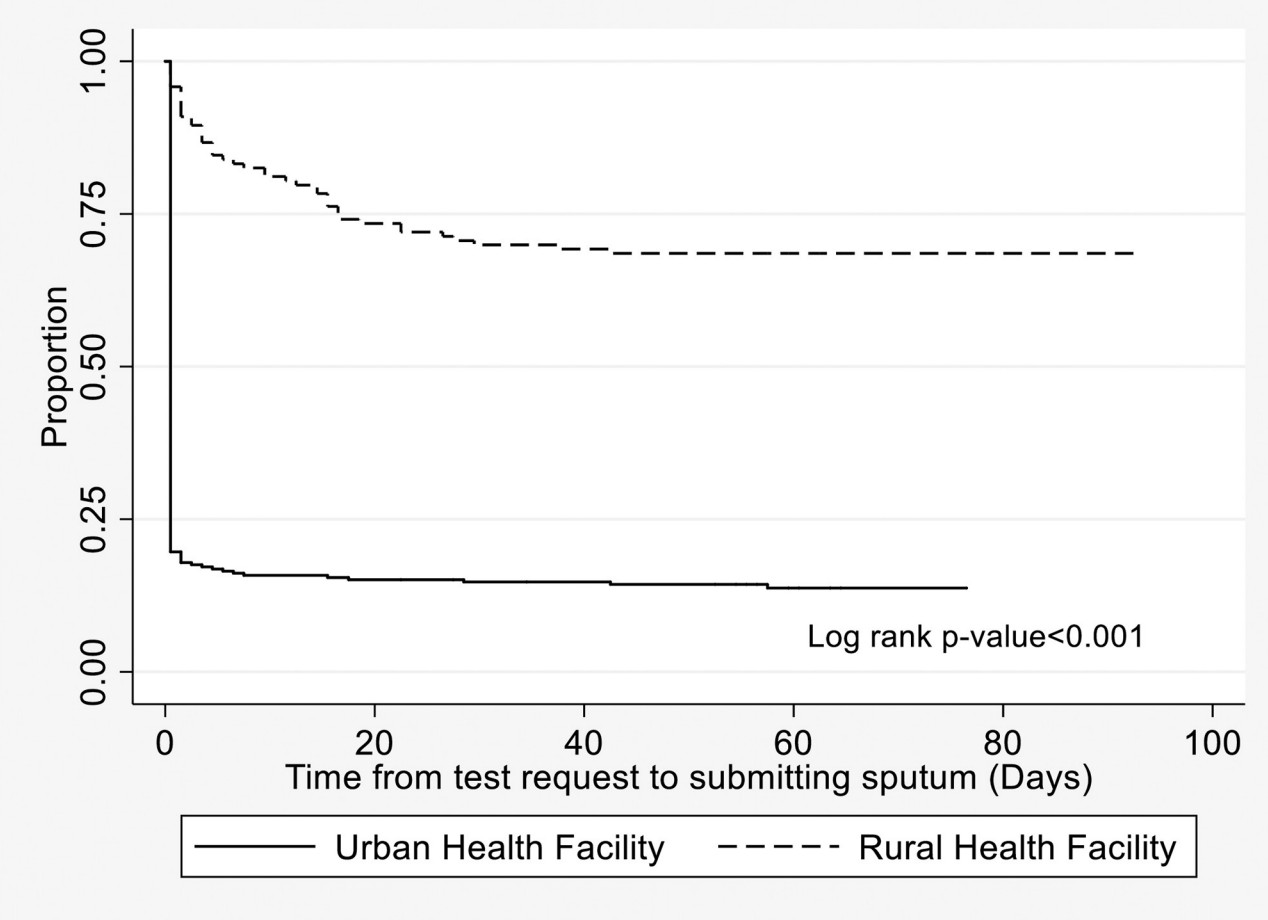

**Fig 3. Time from test request to submitting sputum for people requested to give sputum for TB investigation in Ketu South Municipality, Ghana.**

in Ghana in 2011 where pre-treatment loss-to-follow up was 38% [30], and a systematic review which found an 18% pre-treatment loss to follow-up in Africa [31]. The difference could be due to the presence of a "task-shifting officer", dedicated to TB screening and follow-up in our study hospital's main outpatient department, supporting the value of this role.

**Table 2. Causal analysis showing association between the type of health facility attended and submitting a sputum for a TB test, Ketu South Municipal—Ghana.**

| Association between type of health facility attended and submitting sputum | OR | CI | P |
|---|---|---|---|
| Univariable+ | 24.25 | 13.84–42.51 | P<0.001 |
| Confounder adjustment | | | |
| Adjusted for longer distance from residence to laboratory | 12.98 | 5.95–28.30 | P<0.001 |
| Adjusted for rural residence | 18.25 | 8.08–41.21 | P<0.001 |
| Adjusted for negative HIV status | 22.07 | 12.35–39.44 | P<0.001 |
| Adjusted for high perceived stigma | 22.46 | 12.73–39.61 | P<0.001 |
| Adjusted for increasing number of symptoms | 22.64 | 12.85–39.88 | P<0.001 |
| **Final: adjusted for all of above** | 9.52 | 3.87–23.40 | P<0.001 |

†bivariate analysis between type of health facility (rural or urban) and sputum submission as an outcome.

OR = odds ratio, CI = 95% confidence interval, P = p-value

**Table 3. Factors associated with submitting sputum among study participants attending rural and urban health facilities in Ketu South Municipality, Ghana.**

| Variable | Number submitting sputum (%) | OR (95% CI) | P | aOR (95%CI) | P |
|---|---|---|---|---|---|
| | | N = 410 | | | |
| **Age group (years)** | | | *0.689 | | |
| 18–38 | 99/145 (68.3) | 1 | | | |
| 39–59 | 107/150 (71.3) | 1.15 (0.70–1.90) | 0.567 | | |
| ≥60 | 84/115 (73.0) | 1.26 (0.73–2.16) | 0.403 | | |
| **Gender** | | | *0.177 | | |
| Male | 102/136 (75.0) | 1 | | 1 | |
| Female | 188/274 (68.6) | 0.73 (0.46–1.16) | 0.182 | 0.90 (0.48–1.69) | 0.739 |
| **Educational level** | | | *0.706 | | |
| No formal education | 65/96 (67.7) | 1 | | | |
| Primary/JHS | 162/228 (71.1) | 1.17 (0.70–1.96) | 0.549 | | |
| Secondary/Tertiary | 63/86 (73.3) | 1.31 (0.69–2.48) | 0.414 | | |
| **Place of residence** | | | *<0.001 | | |
| Urban | 182/197 (92.4) | 1 | | 1 | |
| Rural | 108/213 (50.7) | 0.08 (0.05–.015) | <0.001 | 0.41 (0.16–1.03) | 0.058 |
| **Distance to laboratory (Km)** | | | *<0.001 | | |
| <10 | 184/197 (93.4) | 1 | | 1 | |
| 10–20 | 74/168 (44.1) | 0.06 (0.03–0.11) | <0.001 | 0.12 (0.05–0.33) | <0.001 |
| >20 | 32/45 (71.1) | 0.17 (0.07–0.41) | <0.001 | 0.29 (0.10–0.86) | 0.025 |
| **Socioeconomic status (tertiles)** | | | *0.002 | | |
| Low | 92/134 (68.7) | 1 | | 1 | |
| Middle | 86/138 (62.3) | 0.76 (0.46–1.25) | 0.272 | 0.59 (0.30–1.13) | 0.113 |
| High | 112/138 (81.2) | 1.97 (1.12–3.45) | 0.018 | 1.03 (0.47–2.24) | 0.945 |
| **Ever treated for TB** | | | *0.008 | | |
| No | 263/380 (69.2) | 1 | | 1 | |
| Yes | 27/30 (90.0) | 4.00 (1.19–13.46) | 0.025 | 2.02 (0.45–9.07) | 0.359 |
| **Number of symptoms** | | | *<0.001 | | |
| 1–2 | 34/56 (60.7) | 1 | | 1 | |
| 3–4 | 123/191 (64.4) | 1.17 (0.63–2.16) | 0.615 | 1.33 (0.60–2.96) | 0.488 |
| >4 | 133/163 (81.6) | 2.87 (1.47–5.59) | 0.002 | 2.00 (0.83–4.80) | 0.12 |
| **Duration of symptoms (days)** | | | *<0.001 | | |
| ≤14 | 158/245 (64.5) | 1 | | 1 | |
| >14 | 132/165 (80.0) | 2.20 (1.39–3.50) | 0.001 | 0.83 (0.43–1.62) | 0.589 |
| **HIV status** | | | *0.037 | | |
| Positive | 28/28 (100.0) | 1 | | 1 | |
| Negative | 66/85 (77.7) | 1.79 (1.02–3.15) | 0.043 | 0.85 (0.40–1.83) | 0.678 |
| Don't know+ | 196/297 (66.0) | - | - | - | - |
| **Severity of illness (Karnofsky score)** | | | *0.024 | | |
| Less severe (≥90) | 136/217 (62.7) | 1 | | 1 | |
| Moderately severe (70–80) | 133/167 (79.6) | 2.70 (1.24–5.92) | 0.013 | 1.84 (0.68–4.98) | 0.228 |
| Severe (≤60) | 21/26 (80.8) | 0.92 (0.17–5.10) | 0.924 | 0.14 (0.01–2.28) | 0.165 |
| **Number of previous care providers visited** | | | *<0.001 | | |
| 0 | 173/272 (63.6) | 1 | | 1 | |
| ≥1 | 117/138 (84.8) | 3.18 (1.88–5.40) | <0.001 | 2.05 (1.02–4.13) | 0.045 |
| **Stigma** | | | *<0.001 | | |
| Low perception | 134/164 (81.7) | 1 | | 1 | |

(*Continued*)

**Table 3.** (Continued)

| Variable | N = 410 | | | | |
| --- | --- | --- | --- | --- | --- |
| | Number submitting sputum (%) | OR (95% CI) | P | aOR (95%CI) | P |
| High perception | 156/246 (63.4) | 0.39 (0.24–0.62) | <0.001 | 0.54 (0.30–0.98) | 0.043 |

*Log likelihood P-value,

+Omitted from model due to collinearity,

OR = odds ratio, aOR = adjusted odds ratio, P = p-value, JHS (13–15 years) = Junior high school

Mortality among study participants was high, with 3% documented to have died within two months of enrolment, which is high for a population of presumptive TB patients but supports data from Zimbabwe [32]. An additional, 4% of participants recruited were lost to follow-up, if all these had died, two-month mortality would have been 7%. This further emphasizes the need for better access to prompt diagnosis and treatment for TB and other conditions.

Findings from this study have both programmatic and health system implications. This study clearly shows there is a gap in the cascade between a sputum test being requested and sputum being submitted, particularly among persons attending rural health facilities, and this can be attributed to the distance people need to travel to submit a sputum. An effective specimen transport system might bridge the gap in the diagnostic cascade.

## Limitations

The study was designed to be observational rather than interventional and so the follow-up calls aimed to find out whether the participant had given a specimen and/or started TB treatment rather than motivating them to give a specimen; however, follow-up calls could have prompted some participants to give a sputum specimen, and thus we may have underestimated the true pre-diagnostic delays or losses.

## Strengths

The prospective cohort design of our study allowed us to determine pre-diagnostic losses directly and explore the reasons for losses, whereas previous studies have been based on secondary data analysis, or recruited patients already on TB treatment, thus excluding those who never started treatment.

## Conclusion

Almost 30% of patients asked to give a sputum specimen for TB testing in Ketu South municipality, Ghana, did not submit sputum: this was primarily determined by longer distance to the TB diagnostic laboratory. Closing this gap in the TB diagnostic cascade is an important step in reducing treatment delay and reducing TB transmission.

## Acknowledgments

The authors thank the staff of the Municipal Health Directorate, all participating health facilities, research assistants and all study participants for their cooperation and support. Thanks to the Commonwealth Scholarship Commission for granting JD a scholarship for her doctoral studies. The Commission had no involvement in any aspect of the study and manuscript.

## Author Contributions

**Conceptualization:** Joyce B. Der, Alison D. Grant.

**Data curation:** Joyce B. Der, Clement T. Narh.

**Formal analysis:** Joyce B. Der, Daniel Grint, Clement T. Narh.

**Investigation:** Joyce B. Der, Clement T. Narh.

**Methodology:** Joyce B. Der, Daniel Grint, Clement T. Narh, Alison D. Grant.

**Project administration:** Joyce B. Der.

**Supervision:** Daniel Grint, Frank Bonsu, Alison D. Grant.

**Writing – original draft:** Joyce B. Der.

**Writing – review & editing:** Daniel Grint, Clement T. Narh, Frank Bonsu, Alison D. Grant.

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
