## [Decision Letter · Decision Letter 0]

31 Dec 2019

PONE-D-19-34140

Where are patients missed in the tuberculosis diagnostic cascade?  A prospective cohort study in Ghana

PLOS ONE

Dear Prof. Der,

Thank you for submitting your manuscript to PLOS ONE. After careful consideration, we feel that it has merit but does not fully meet PLOS ONE’s publication criteria as it currently stands. Therefore, we invite you to submit a revised version of the manuscript that addresses the points raised during the review process.

We would appreciate receiving your revised manuscript by Feb 14 2020 11:59PM. To enhance the reproducibility of your results, we recommend that if applicable you deposit your laboratory protocols in protocols.io, where a protocol can be assigned its own identifier (DOI) such that it can be cited independently in the future. For instructions see: http://journals.plos.org/plosone/s/submission-guidelines#loc-laboratory-protocols

We look forward to receiving your revised manuscript.

Kind regards,

HASNAIN SEYED EHTESHAM

Academic Editor

PLOS ONE

Journal Requirements:

2. Please include additional information regarding the survey or questionnaire used in the study and ensure that you have provided sufficient details that others could replicate the analyses. For instance, if you developed a questionnaire as part of this study and it is not under a copyright more restrictive than CC-BY, please include a copy, in both the original language and English, as Supporting Information. In addition, please provide any details of pretesting of the questionnaire, if such pretesting took place, including the number of participants and where they were recruited from.

Additional Editor Comments:

I have gone through this manuscript and also the comments of both the Reviewers. The Authors may need to address all comments raised by both the reviewers to add more value to this manuscript. My decision is Minor revision.

Reviewers' comments:

Reviewer's Responses to Questions

**Comments to the Author**

1. Is the manuscript technically sound, and do the data support the conclusions?

Reviewer #1: Yes

Reviewer #2: Yes

2. Has the statistical analysis been performed appropriately and rigorously? 

Reviewer #1: I Don't Know

Reviewer #2: Yes

3. Have the authors made all data underlying the findings in their manuscript fully available?

Reviewer #1: Yes

Reviewer #2: Yes

4. Is the manuscript presented in an intelligible fashion and written in standard English?

Reviewer #1: Yes

Reviewer #2: Yes

5. Review Comments to the Author

Reviewer #1: Recording of positive family history of TB is not mentioned. if that is available in the records that should be mentioned .

Also if TB was suspected as a complication of HIV has not been recorded although presence / absence of HIV has been noted.

Reviewer #2: Comments to the authors:

The current study entitled “Where are patients missed in the tuberculosis diagnostic cascade? A prospective cohort study in Ghana” by Joyce B. Der et al. reported the possible reasons and to circumvents these in lower TB diagnosis rate in Ghana. They showed in the current study that distance of the diagnostic centers from the reporting center is the main hurdle to submit the sample for diagnosis. Although, the population study size chosen is moderate as well as the age and gender is somewhat biased. These could be managed at the time of patient recruitment. The study well is designed, executed and presented. Data are presented in legible manner and are easily understandable. Manuscript is well written. The authors tried to find the reasons behind the less reporting of the TB cases in a region of Ghana. They determined the casual link and suggested the measures to combat the problems. The information provided in the manuscript can be used to fill the gap that exist in the regions of Ghana and if followed may increase the cases of TB reporting and their diagnosis. These information’s can also be used to increase the rate of TB diagnosis and its management in high TB burden areas of the Ghana. The manuscript is suitable and could be considered for publication in PLOS ONE.

6. PLOS authors have the option to publish the peer review history of their article (what does this mean?). If published, this will include your full peer review and any attached files.

Reviewer #1: No

Reviewer #2: Yes: Mohd Shariq

---

## [Author Response · Author response to Decision Letter 0]

15 Feb 2020

PONE-D-19-34140: Where are patients missed in the tuberculosis diagnostic cascade? A prospective cohort study in Ghana: Der J et al 

We thank the reviewers for their positive feedback and comments. Our responses to the comments are below:

Reviewers’ comments:

Reviewer #1

Comment: Recording of positive family history of TB is not mentioned. if that is available in the records that should be mentioned.

Response: Thank you for this comment. Unfortunately we did not collect information on positive family history of TB and therefore cannot include it.

Comment: Also if TB was suspected as a complication of HIV has not been recorded although presence / absence of HIV has been noted

Response: Thank you for the comment. We recruited people who were attending the general outpatients’ department using a referral system through the task shifting officer who is responsible for screening people with symptoms of TB and referring them to the laboratory for a sputum test. People attending for HIV care who needed investigation for TB were referred through this system and some will have been included in our study. We asked participants to report their HIV status if they knew it, and as you say we have reported this. However we did not ask participants their route of referral and so we are unable to report how many participants were referred from the antiretroviral (ART) clinic to the task shifting officer. 

Reviewer #2

Comment: The current study entitled “Where are patients missed in the tuberculosis diagnostic cascade? A prospective cohort study in Ghana” by Joyce B. Der et al. reported the possible reasons and to circumvents these in lower TB diagnosis rate in Ghana. They showed in the current study that distance of the diagnostic centers from the reporting center is the main hurdle to submit the sample for diagnosis. Although, the population study size chosen is moderate as well as the age and gender is somewhat biased. These could be managed at the time of patient recruitment. The study well is designed, executed and presented. Data are presented in legible manner and are easily understandable. Manuscript is well written. The authors tried to find the reasons behind the less reporting of the TB cases in a region of Ghana. They determined the casual link and suggested the measures to combat the problems. The information provided in the manuscript can be used to fill the gap that exist in the regions of Ghana and if followed may increase the cases of TB reporting and their diagnosis. These information’s can also be used to increase the rate of TB diagnosis and its management in high TB burden areas of the Ghana. The manuscript is suitable and could be considered for publication in PLOS ONE.

Response: Thank you for the positive feedback about our study. In relation to the comment on age and sex being somewhat biased, our study was designed to consecutively select patients with symptoms of TB who have been routinely asked by a health worker to do a sputum test. We believe the study participants were therefore representative of people being asked to give a sputum for a TB test. We respectfully disagree with the comment. We have not changed the manuscript.

Editorial points to be addressed:

Response: This has been done

Please include additional information regarding the survey or questionnaire used in the study and ensure that you have provided sufficient details that others could replicate the analyses. For instance, if you developed a questionnaire as part of this study and it is not under a copyright more restrictive than CC-BY, please include a copy, in both the original language and English, as Supporting Information. In addition, please provide any details of pretesting of the questionnaire, if such pretesting took place, including the number of participants and where they were recruited from.

Response: The questionnaires developed for this study will be deposited with London School of Hygiene and Tropical Medicine’s (LSHTM) Data Compass and a link provided so anyone interested can have unrestricted access.

The questionnaire was pretested at the municipal hospital among adults aged 18 years or more routinely identified by a health worker as requiring a sputum TB test. A total of eight participants were recruited, 4 (50%) males and 4 (50%) females aged between 27-85 years (mean age: 55 year [SD:18]). 

We have added two lines on pretesting of the questionnaire in the manuscript (line 117-120 on page 5) which read:

The questionnaire was pretested at the municipal hospital in a population similar to the study population. Following pretesting, revisions were made to improve comprehensibility and understanding.

We note that you have stated that you will provide repository information for your data at acceptance. Should your manuscript be accepted for publication, we will hold it until you provide the relevant accession numbers or DOIs necessary to access your data. If you wish to make changes to your Data Availability statement, please describe these changes in your cover letter and we will update your Data Availability statement to reflect the information you provide.

Response: We wish to maintain that data will be deposited with the London School of Hygiene and Tropical Medicine’s Data Compass. However, due to the presence of sensitive patient information in the dataset, access will be restricted. It is recommended that persons who wish to access the dataset should click on the DOI link and then click on the 'request access' button to fill a form which will be automatically submitted to the research data manager at London School of Hygiene and Tropical Medicine and to the first author. However, a non-author email address for request of the dataset is researchdatamanagement@lshtm.ac.uk.

---

## [Editor Report · Decision Letter 1]

4 Mar 2020

Where are patients missed in the tuberculosis diagnostic cascade?  A prospective cohort study in Ghana

PONE-D-19-34140R1

Dear Dr. Der,

We are pleased to inform you that your manuscript has been judged scientifically suitable for publication and will be formally accepted for publication once it complies with all outstanding technical requirements.

With kind regards,

HASNAIN SEYED EHTESHAM

Academic Editor

PLOS ONE

Additional Editor Comments (optional):

This manuscript was reviewed by 2 reviewers who were experts in the field. One of the reviewers has asked questions about family history of TB which the Authors were unable to provide since this was not in their record and was not designed to be part of this study. The Reviewer 2 also had a small issue about age and sex information which was also not part of the original study. The issues about the questionnaire for the study has been addressed. The other important issue about the repository and deposition of data has been clarified and these will be deposited in the London School of Hygiene and Tropical Medicine’s data compass. The manuscript is now recommended for publication.
---

## [Editor Report · Acceptance letter]

6 Mar 2020

PONE-D-19-34140R1 

Where are patients missed in the tuberculosis diagnostic cascade?  A prospective cohort study in Ghana 

Dear Dr. Der:

I am pleased to inform you that your manuscript has been deemed suitable for publication in PLOS ONE. Congratulations! Your manuscript is now with our production department. 

With kind regards,

on behalf of

Prof HASNAIN SEYED EHTESHAM 

Academic Editor

PLOS ONE